# Preliminary Analysis of the Effects of Ad26.COV2.S Vaccination on CT Findings and High Intensive Care Admission Rates of COVID-19 Patients

Davide Negroni [1,*] , Serena Carriero [2], Ilaria Passarella [3], Agnese Siani [1], Pierpaolo Biondetti [2], Antonio Pizzolante [1], Luca Saba [4] and Giuseppe Guzzardi [1]

1 Department of Radiology, University of Piemonte Orientale, Piedmont, 28100 Novara, Italy
2 Department of Radiology, Fondazione IRCCS Ca' Granda Ospedale Maggiore Policlinico, 20122 Milan, Italy
3 Departement of Internal Medicine, ASST Ovest Milanese Ospedale Fornaroli, 20013 Magenta, Italy
4 Department of Radiology, Azienda Ospedaliero Universitaria Cagliari, 09124 Polo di Monserrato, Italy
* Correspondence: dvdngr@gmail.com; Tel.: +39-3348163299

**Abstract:** On 27 February 2021, the Food and Drug Administration(FDA) authorized the administration of the adenovirus-based Ad26.COV2-S vaccine (J&J-Janssen) for the prevention of COVID-19, a viral pandemic that, to date, has killed more than 5.5 million people. Performed during the early phase of the COVID-19 4th wave, this retrospective observational study aims to report the computerized tomography (CT) findings and intensive care unit admission rates of Ad26.COV2-S-vaccinated vs. unvaccinated COVID-19 patients. From the 1st to the 23rd of December 2021, all confirmed COVID-19 patients that had been subjected to chest non-contrast CT scan analysis were enrolled in the study. These were divided into Ad26.COV2.S-vaccinated (group 1) and unvaccinated patients (group 2). The RSNA severity score was calculated for each patient and correlated to CT findings and type of admission to a healthcare setting after CT—i.e., home care, ordinary hospitalization, sub-intensive care, and intensive care. Descriptive and inference statistical analyses were performed by comparing the data from the two groups. Data from a total of 71 patients were collected: 10 patients in group 1 (4M, 6F, mean age 63.5 years, SD ± 4.2) and 61 patients in group 2 (32M, 29F, mean age 64.7 years, SD ± 3.7). Statistical analysis showed lower values of RSNA severity in group 1 compared to group 2 (mean value 14.1 vs. 15.7, $p = 0.009$, respectively). Furthermore, vaccinated patients were less frequently admitted to both sub-intensive and high-intensive care units than group 2, with an odds ratio of 0.45 [95%CI (0.01; 3.92)]. Ad26.COV2.S vaccination protects from severe COVID-19 based on CT severity scores. As a result, Ad26.COV2.S-vaccinated COVID-19 patients are more frequently admitted to home in comparison with unvaccinated patients.

**Keywords:** COV2-S vaccine; COVID-19 disease; computerized tomography; RSNA severity

## 1. Introduction

Since the first reported case of coronavirus disease 2019 (COVID-19) was reported in the city of Wuhan in December 2019 [1], severe acute respiratory syndrome coronavirus 2 (SARS-CoV-2) infections have spread worldwide. This has led the World Health Organization (WHO) to declare the COVID-19 outbreak a global pandemic on 11 March 2020 [2], which to date has caused more than 5.5 million deaths. As 14% of COVID-19 patients experience a severe form of the disease and 5% of them require admission to intensive care units (ICUs) [3], virtually all national health systems worldwide have been put under unprecedented strain.

Chest high-resolution computed tomography (HRCT) is the gold-standard radiological technique for the detection of SARS-CoV-2-associated pneumonia, with a sensitivity of 97%, specificity of 25%, and accuracy of 68% in comparison with RT-PCR [4]. The advantages brought by chest HRCT in COVID-19 diagnosis are mainly ascribable to its ability to

recognize several characteristic patterns, such as ground glass opacity (GGO), consolidation, crazy-paving, and, less frequently, reverse halo-sign [5,6]. In this regard, several studies have shown that reduced aerated lung volume and increased GGO and/or consolidation and fibrosis are indicators of poor outcome in COVID-19 patients. Thus, chest CT scan represents a helpful radiological modality in the evaluation and management of SARS-CoV-2-infected patients [7].

However, in the daily work of a radiologist during a pandemic, the segmentation of the lung parenchyma of COVID-19 patients, necessary to conduct accurate CT-based diagnoses, is hardly ever available. Hence, the need for a new algorithm that allows for evaluating the severity of pulmonary involvement quickly and objectively. In this regard, the RSNA CT severity score (RSNA CT-SS) has been recently proposed as a useful tool to investigate pulmonary COVID-19 severity [8]. To perform RSNA CT-SS analysis, both lungs are divided into 18 segments, according to their anatomical structure. Moreover, as described by Yang et al. [8], "The posterior apical segment of the left upper lobe" is" subdivided into apical and posterior segmental regions, whereas the anteromedial basal segment of the left lower lobe" is "subdivided into anterior and basal segmental regions", thus obtaining 20 regions. Each region is then evaluated by the radiologist on chest HRCT, attributing scores of 0, 1, or 2 if the parenchymal opacification involves, respectively, 0%, <50%, or ≥50% of each region. The theoretic RSNA CT-SS score is calculated as the sum of the individual scores in the 20 lung segment regions, ranging from 0 to 40 points.

The Ad26.COV2-S vaccine (J&J-Janssen) is a recombinant, replication-defective human adenovirus type 26 vector encoding the full-length spike protein of SARS-CoV-2 [9]. Since its approval for human use by the Food and Drug Administration (FDA) in February 2021, about 30,000,000 of dose have been administered in the United States [10]. As of 15 January 2022, the prevalence of the Ad26.COV2-S vaccine was about 9% among 60- to 70-year-old subjects [11]. In this regard, a study by Rosenberg et al. has shown that Ad26.COV2-S vaccine effectiveness (VE) against hospitalization was of 80.4 (71.9 to 86.7) (95% CI) in a cohort of COVID-19 patients aged 65 years or older after at least 15 days from the vaccine [9]. However, there is paucity of studies on hospitalized Ad26.COV2-S-vaccinated COVID-19 patients.

This study aimed to describe the effects of Ad26.COV2-S vaccination on CT findings and hospitalization rates among COVID-19 patients using the RSNA CT severity score.

## 2. Materials and Methods

### 2.1. Patients

This retrospective study was approved by our Institutional Review Boards (number CE 130/20). All patients who had undergone chest HRCT imaging between the 1st and the 23rd of December 2021 were sequentially included in the study. The inclusion criteria were: (1) age between 60 and 70 years old; (2) a COVID-19 positive swab (molecular or rapid swab) within the last 24 h before the examination. The exclusion criteria were: (1) vaccination with another vaccine than Ad26.COV2-S or heterogeneous vaccination, (2) vaccination with Ad26.COV2-S at less than 28 days, and (3) COVID-19 Reporting and Data System (CO-RADS) < 3 [12].

The CO-RADS is a scheme used to standardize the assessment of COVID-19 pulmonary involvement on chest CT findings; only CO-RADS of 3, 4, and 5 (3, uncertain; 4, high probability; 5, very high probability, respectively) were considered for this study [12]. All CO-RADS were subsequently confirmed by nose-pharyngeal swab.

Ad26.COV2-S-vaccinated patients enrolled in the study were included in group 1, while all the unvaccinated ones were considered as group 2.

After chest HRCT, the patients' demographic data (i.e., sex and age) and admission rates to healthcare settings were collected.

The different types of admission to healthcare settings being considered were the following: Home Care; Ordinary Hospitalization; Subintensive Care; Intensive Care; and High Intensity Hospitalization (i.e., Subintensive Care + Intensive Care).

### 2.2. Chest HRCT Acquisition and Examination

The hospital protocol planned to perform a chest HRCT on all patients with moderate-to-severe COVID-19 symptoms (dyspnea, thorax pain) and oxygen saturation < 92% or P/F ratio < 350 or in respiratory acidosis.

All images collected were anonymized by a study researcher (D.N.). Two expert radiologists—each having performed more than 2000 chest CTs on COVID-19 patients—analyzed retrospectively the imaging. The agreement between the two radiologists on CO-RADS, RSNA CT severity scores was 0.86. All HRCTs with CO-RADS values < 3 were excluded from the study. The evaluating radiologists were blinded to SARS-CoV-2 positivity.

The following chest HRCT protocol was used for acquisition: tube voltage: 120 kV; tube current modulation: 226 mAs; spiral pitch factor: 1.08; collimation width 0.625, matrix 512 (mediastinal window) and 768 (lung window). All images were reconstructed with a 1 mm slice thickness range using both sharp kernels (B70f) with a standard lung window (1500 width; −500 centers) and medium-soft kernels (B40f) with a soft-tissue window (300 widths; 40 centers). The images in Digital Imaging and Communications in Medicine (DICOM) extension files were transferred to the Picture Archiving and Communication System (PACS) of our institution and then analyzed into a workstation equipped with two 35 × 43-cm monitors (produced by Eizo, with 2048 × 1536 matrix).

All chest HRCT scans were performed on a 128-slice CT scanner (Philips Ingenuity Core, Philips Healthcare, Netherlands) in the supine position during a single full inspiratory breath-hold.

### 2.3. Statistical Analysis

Descriptive analyses were conducted on the patient's data enrolled in the study. The categorical variables were synthesized by absolute and relative frequencies, while the numerical ones were analyzed through means and standard deviation (SD). To evaluate significant differences concerning sociodemographic and clinical variables, parametric, non-parametric tests, and odds ratios (OR) were performed. The significance level was set at 0.05, and the analyses were conducted using STATA 13.0 software, College Station, TX: StataCorpLLC, Texas.

### 3. Results

A total of 71 out of 185 patients were enrolled in the study. Among them, 10/71 (14.1%) had received a single dose of Ad26.COV2-S (group 1), which made them fully vaccinated, while 61/71 (85.9%) had not received any type of vaccination (group 2). The mean age of our sample (36M and 35F) was 64.5 years SD ± 3.8, with a mean RSNA severity scale of 15.2 SD ± 10.1. Group 1 was vaccinated on average 102 SD ± 15 days after admission to the emergency radiology department. Group 1 and group 2 patients' characteristics are summarized in Table 1. None of the patients had double access to the Emergency Radiology in the study period. The 80% (8/10) of the vaccinated patients (group 1) were treated with Home Care, while 10% (1/10) were sent to a Hight Intensity Hospitalization department (understood as Subintensive Care + Intensive Care).

Among the non-vaccinated patients (group 2), 54.1% (33/61) were admitted to a Home Care; 19.7% (12/61) of group 2 were admitted to a Hight Intensity Hospitalization department.

Univariate statistical analysis on the type of admission between group 1 and group 2 did not identify statistically significant differences ($p > 0.05$).

The OR to calculate the post-vaccination risk of being admitted to Home Care, Ordinary Hospitalization, Subintensive Care, Intensive Care, and High Intensity Hospitalization were, respectively: 3.39 [adjusted 95% CI, 0.60 to 34.83], 0.31 [adjusted 95% CI, 0.07 to 2.62], 0.74 [adjusted 95% CI, 0.02 to 6.80], 0.0 [adjusted 95% CI, 0.00 to 6.07], and 0.45 [adjusted 95% CI, 0.01 to 3.92].

**Table 1.** It shows the main characteristics of the study sample, represented by mean with standard deviation, absolute number, frequency with the respective *p*-value. Group 1: Ad26.COV2-S vaccinated; Group 2: non-vaccinated; RSNA SS: RSNA severity score; HC: Home Care; OH: Ordinary Hospitalization; SC: Subintensive Care; IC: Intensive Care; HI: High Intensity Care (SC+IC); SD: Standard Deviation.

|  | **Group 1** | **Group 2** |  |
|---|---|---|---|
|  | Mean ± SD | Mean ± SD | *p* |
| Age | 63.5 ± 4.2 | 64.7 ± 3.7 | 0.178 |
| RSNA SS | 9.7 ± 4.4 | 16.2 ± 10.5 | 0.030 |
|  | N% | N% |  |
| TOT | 10 (100%) | 61 (100%) |  |
| Male | 4 (40%) | 32 (52.5%) | 0.465 |
| Female | 6 (60%) | 29 (47.5%) |  |
| Admission to healthcare settings |  |  |  |
| Home Care | 8 (80%) | 33 (54.1%) | 0.124 |
| Ordinary Hospitalization | 1 (10%) | 16 (26.2%) | 0.265 |
| Subintensive Care | 1 (10%) | 8 (13.1%) | 0.784 |
| Intensive Care | 0 (0.0%) | 4 (6.6%) | 0.405 |
| High Intensity Hospitalization | 1 (10%) | 12 (19.7%) | 0.464 |

The RSNA CT-SS, sex, and age sorted by hospitalization type are summarized in Table 2. The statistical analyses did not report any difference from Table 2 sorted data (*p* > 0.05). The RSNA CT-SS were similar between males and females, with mean values of 16.2 SD ± 10.1 and 14.3 SD ± 10.1 for M and F (*p* = 0.21), respectively.

**Table 2.** It shows CT, gender, and sex characteristics of patients grouped by type of admission; the data are represented by mean with standard deviation, absolute number and frequency. RSNA SS: RSNA severity score; High Intensity Hospitalization (Subintensive Care + Intensive Care); SD: Standard Deviation.

| Admission to Healthcare Settings | TOT | Male | Female | Age ± SD | RSNA SS ± SD |
|---|---|---|---|---|---|
| Home Care | 41 (100%) | 18 (43.9%) | 23 (56.1%) | 63.8 ± 3.4 | 8.0 ± 3.6 |
| Ordinary Hospitalization | 17 (100%) | 11 (64.7%) | 6 (35.3%) | 65.2 ± 4.3 | 22.2 ± 6.7 |
| Subintensive Care | 9 (100%) | 6 (66.7%) | 3 (33.3%) | 65.3 ± 4.4 | 28.0 ± 6.6 |
| Intensive Care | 4 (100%) | 1 (25.0%) | 3 (75.0%) | 67.7 ± 2.6 | 31.3 ± 4.9 |
| High Intensity Hospitalization | 13 (100%) | 7 (53.8%) | 6 (46.2%) | 66.1 ± 4.1 | 29.0 ± 6.1 |

The RSNA CT-SS mean value analysis showed a statistical difference between the hospitalization type (*p* < 0.001). In detail, patients hospitalized in High Intensity units presented a mean RSNA CT-SS of 29.0 SD ± 6.1, in Ordinary Hospitalization 22.2 SD ± 6.7, and in Home Care 8.0 SD ± 3.6. The mean difference between High Intensity Hospitalization and Ordinary Hospitalization in RSNA CT-SS score was 7 points, whereas, between High Intensity Hospitalization and Home Care, the mean difference was 14 points.

## 4. Discussion

This study aimed to evaluate the possible effect of Ad26.COV2-S vaccination vs. no vaccination on chest HRCT evaluation and subsequent admission to healthcare facilities of COVID-19 patients.

Chest HRCT of Ad26.COV2-S-vaccinated subjects had statistically significant lower RNSA CT-SS values than those recorded in unvaccinated subjects (Figures 1 and 2). This finding correlated with the ability of the Ad26.COV2-S vaccine to reduce severe-critical COVID-19. Specifically, for patients with onset at ≥14 days, the risk of severe disease was

76.7% [adjusted 95% CI, 54.6 to 89.1], whereas, for those with onset at ≥28 days, it was 85.4% [adjusted 95% CI, 54.2 to 96.9], as reported from Sandoff and colleagues [9].

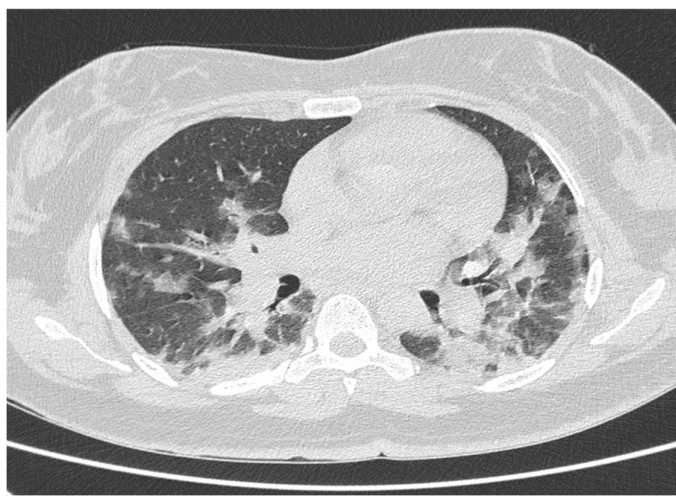

**Figure 1.** CT scan of a 66-year-old unvaccinated woman with COVID-19 interstitial pneumonia. CO-RADS 4, confirmed by molecular swab; RNSA CT severity score of 22 points. The patient was admitted to the subintensive care unit.

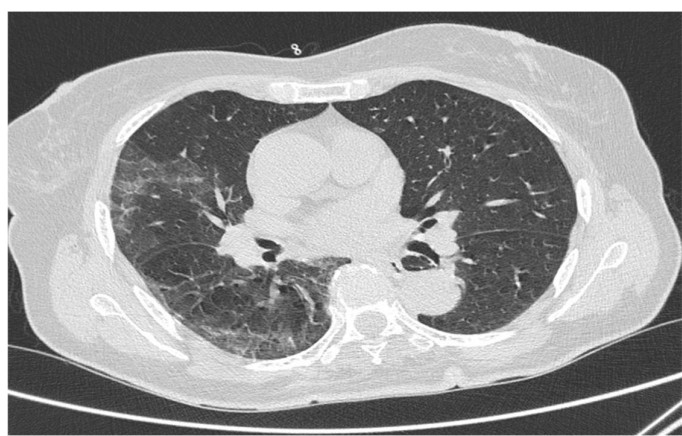

**Figure 2.** Ct scan of a 62-year-old woman vaccinated with Ad26.COV2.S suffering from COVID-19 interstitial pneumonia. CO-RADS 3 was estimated (confirmed by molecular swab), with an RSNA CT severity score of 12 points. The patient was admitted to ordinary care.

Thus, the Ad26.COV2-S vaccine seems to be able to reduce the presentation of COVID-19 from a clinical point of view and also to exert a protective role against viral pneumonia.

The ratio of hospitalizations between the vaccinated and the non-vaccinated was 17%, which is lower than the current scientific literature, which estimates the rate of hospitalisations among those vaccinated with Ad26.COV2-S at between 25% and 40% [13–16]. This discrepant figure could be due to the strict restrictions that have been applied in Italy aimed at containing the spread of the pandemic.

Our findings are also supportive of a protective role of Ad26.COV2-S vaccination against Ordinary Hospitalization, Subintensive Care and Intensive Care (OR < 0.5). In particular, the possibility of being admitted to a high intensity care unit (Subintensive Care or Intensive Care) was reduced by 55% in Ad26.COV2-S-vaccinated COVID-19 patients. Furthermore, the likelihood that Ad26.COV2-S-vaccinated patients could avoid hospitalization and be managed at home increased by more than three times compared to unvaccinated patients. More specifically, around 80% of Ad26.COV2-S-vaccinated patients were admitted

to Home Care vs. 54.1% of unvaccinated ones. This result may be ascribable to the lower CT severity of COVID-19 lung impairment observed in vaccinated vs. unvaccinated patients.

The protection against the development of more severe forms of COVID-19 could be explained by the immune re-response acquired by Ad26.COV2-S vaccination. In unvaccinated subjects, a response develops based on T lymphocytes, macrophages and IFNs attacking infected alveolar cells, with high production of IL-6 and pro-inflammatory cytokines [16,17].

Subjects vaccinated with Ad26.COV2-S develop B cells capable of attacking the spike protein present on the outer surface of Sars.Cov.2. The spike protein is used by the virus to enter human cells. Therefore, the antiviral response mediated by the B lymphocytes reduces the activation of the innate immune system [17,18]. This mechanism protects against virus persistence and the development of lung damage caused by inflammatory hyperactivation.

Group 1 was vaccinated with Ad26.COV2-S between June and August 2021. Early studies on the vaccine's efficacy seemed to demonstrate increased protection with time, reaching a peak of 92.4% at 42 days towards severe forms [9]. Polinski et al. demonstrated the high durability of the Ad26.COV2-S vaccine at 180 days: the study re-estimated an efficacy of 81% [adjusted 95% CI, 78% to 82%] for COVID-19-related hospitalisations [13]. Unfortunately, data about the pre-infection serology status of group 1 subjects are not available; the possibility that they are poor vaccine responders is possible, but the RSNA CT-SS figure suggests that protective immunity is nonetheless present in these patients.

Another possible explanation of our findings is that, in December 2021, the B.1.1.529 variant (Omicron) was reported in multiple countries [19], displaying a much higher vaccine breakthrough rate (88%) compared to other variants of concern (VOCs) and the original Wuhan strain and with a tendency to cause upper rather than lower tract infections—in this regard, studies on Ad26.COV2-S vaccine protection form Omicron are currently scarce [19]. Even though the poor SARS-CoV-2 genomic surveillance system implemented by the Italian Government failed to provide a precise picture of the Omicron prevalence in Italy for that period, the fact that the Omicron was first reported in South Africa on 23 November 2021 [16,17], and that our study was carried out between the 1st and the 23rd of December 2021, should rule out this confounder bias. Importantly, we observed a combined Subintensive Care/Intensive Care hospitalization rate of 10% for group 1 vs. 19.7% for group 2. In contrast to previous studies, we show that vaccinated women were more likely to be admitted to Intensive Care compared to men [20–23]. In particular, Cau et al. [17] found that male patients with a mean age of 60 years SD±11 were those more likely (80%) to develop severe COVID-19 pneumonia compared to same age females. These conflicting findings may be due to the stochastic effect in a limited sample in our study of 71 patients, with only four Intensive Care admissions.

This study has several limitations. Group 1 was limited by the small number of Ad26.COV2-S vaccinated patients; a wider sample, with a mid-term follow-up (e.g., 3 months or 6 months), may ensure a higher accuracy. Furthermore, the sample used is a very select one and the risk of bias is high; group 2 consists of patients who are difficult to identify (vaccinated with Ad26.COV2-S who nevertheless developed clinically relevant pneumonia), limiting the analysis of the data and the evidence grade. The sample came from a population who had access to a single Emergency Radiology. A multicenter study comparing different populations may provide more generalizable results. Finally, the data on patients' clinical status and COVID-19 oxygenation treatments were lacking.

## 5. Conclusions

In this limited study, the Ad26.COV2-S vaccine had a protective role against severe COVID-19 development in patients between the age of 60 and 70 years. In particular, vaccinated people contracted COVID19 pneumonia with a lower degree of severity than the non-vaccinated group (evaluated with RSNA CT Severity Score). Although there was no statistically significant difference, vaccinated COVID-19 patients could be more frequently treated at home, whereas an unvaccinated patient was more likely to be admitted to

Intensive Care or High Intensity units. Larger studies are necessary to confirm the findings. Future studies could also analyse populations subjected to different vaccines to identify any discrepancies in the development of Sars-CoV2 interstitial pneumonia.

**Author Contributions:** Conceptualization, D.N. and I.P.; methodology, D.N. and S.C.; software, D.N.; validation, A.S. and P.B. formal analysis, S.C.; investigation, A.S.; resources, A.P.; data curation, D.N. and A.P.; writing—original draft preparation, D.N. and G.G.; writing—review and editing, A.S.; visualization, S.C.; supervision, G.G. and L.S.; project administration, L.S. and G.G. All authors have read and agreed to the published version of the manuscript.

**Funding:** This research received no external funding.

**Institutional Review Board Statement:** The study was conducted in accordance with the Declaration of Helsinki, and approved by the Institutional Review Board (or Ethics Committee) of AOU Maggiore della Carità di Novara (protocol code CE 130/20; May 2020).

**Informed Consent Statement:** Patient consent was waived due to the retrospective nature of this study.

**Data Availability Statement:** Not applicable.

**Acknowledgments:** I, Davide Negroni, would like to thank Massimiliano Cernigliaro and Francesca Frattini for the moral support and for the useful advice in the text drafting.

**Conflicts of Interest:** The main author, D.N., and the other authors S.C., I.P., A.S., P.B., A.P., L.S. and G.G. declare that they have no conflict of interest.

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
