# Peer review of "Preliminary Analysis of the Effects of Ad26.COV2.S Vaccination on CT Findings and High Intensive Care Admission Rates of COVID-19 Patients"

_tomography, doi:10.3390/tomography8050199_

Round 1

Reviewer 1 Report

The manuscript is very well written and it deals with a very topical subject.

The study is well designed. The study and control group are appropriate.

Though, there are some things that need to be corrected.

Be careful with the descriptions of the figures: figure 1 – it should start – ‘CT scan of… ‘ figure 2 - , ‘this is a figure’ should be delated, again ‘CT scan of’

If possible conclusions should be longer. Maybe some future studies or o bigger nuber of patients should be considered?

  Maybe consider the influence of the time of vaccine intake  previous to the COVID-19 infection on the severity of CT findings. Whether or not the immunity development plays a role in the whole process.

Author Response

The manuscript is very well written, and it deals with a very topical subject.

The study is well designed. The study and control group are appropriate.

Though, there are some things that need to be corrected.

Be careful with the descriptions of the figures: figure 1 – it should start – ‘CT scan of… ‘ figure 2 - , ‘this is a figure’ should be delated, again ‘CT scan of’

If possible, conclusions should be longer. Maybe some future studies or o bigger number of patients should be considered?

"Maybe consider the influence of the time of vaccine intake  previous to the COVID-19 infection on the severity of CT findings. Whether or not the immunity development plays a role in the whole process."

R: I thank the reviewer for his work and advice.
I revised the tables and captions. I have also revised the conclusions, as indicated.

A paragraph on the importance of immunity and its possible role in reducing RSNA CT-SS was included in discussion.

Reviewer 2 Report

COVID-19 pandemic has lasted for also most three years and resulted severe consequences. The authors present the clinical study result of Ad26.COV2-S vaccine (J&J-Janssen) for the prevention of COVID-19. The result suggested the vaccinated patients are more likely to be admitted to home in comparison with unvaccinated patients. This study is important and valuable. The way of presenting needs to be improved. We recommend publication after major revision.

1.     This included relatively small numbers of patients, which makes some of the conclusions statistically nonsignificant (P > 0.05).

2.     The authors use too many non-standard abbreviations (e.g. TOT, HC, OH, SC, HI) in the manuscript, which makes the manuscript difficult to follow.

3.     Table 1 and Table 2 should be redesigned. There are unclear and difficult to follow.

Author Response

COVID-19 pandemic has lasted for also most three years and resulted severe consequences. The authors present the clinical study result of Ad26.COV2-S vaccine (J&J-Janssen) for the prevention of COVID-19. The result suggested the vaccinated patients are more likely to be admitted to home in comparison with unvaccinated patients. This study is important and valuable. The way of presenting needs to be improved. We recommend publication after major revision.

  1. This included relatively small numbers of patients, which makes some of the conclusions statistically nonsignificant (P > 0.05).
  2. The authors use too many non-standard abbreviations (e.g. TOT, HC, OH, SC, HI) in the manuscript, which makes the manuscript difficult to follow.
  3. Table 1 and Table 2 should be redesigned. There are unclear and difficult to follow.

R: I thank reviewer 2 for the careful peer review of my work.

  1. I enriched the conclusions, indicating the limitations of the study.
  2. I expanded the non-standard acronyms to make them easier to read.
  3. I tried to make the tables smoother by changing the representation of the data

Reviewer 3 Report

This article investigates performed during the early phase of the COVID-19 4th wave, this retrospective observational study aims to report the CT findings and intensive care unit admission rates of Ad26.COV2-S-vaccinated vs. unvaccinated COVID-19 patients.

The article is clear and well written, and the authors prove to be skilled in the subject. Citations from the scientific literature are modern, appropriate and adequately integrated into the manuscript. The manuscript standards are adequate with those of "tomography". The manuscript will have a medium impact, in my opinion.

No serious issues are to be reported; hence I recommend the paper for publication after some minor comments.

-The results and discussion section is very poor. I suggest the authors to increase it. The authors need to add more comparison with statistical analyzes.

-Improve the size of Table 2 and 3.

-Please reconfirm the description of Figure 2.

Author Response

This article investigates performed during the early phase of the COVID-19 4th wave, this retrospective observational study aims to report the CT findings and intensive care unit admission rates of Ad26.COV2-S-vaccinated vs. unvaccinated COVID-19 patients.

The article is clear and well written, and the authors prove to be skilled in the subject. Citations from the scientific literature are modern, appropriate, and adequately integrated into the manuscript. The manuscript standards are adequate with those of "tomography". The manuscript will have a medium impact, in my opinion.

No serious issues are to be reported; hence I recommend the paper for publication after some minor comments.

1-The results and discussion section is very poor. I suggest the authors to increase it. The authors need to add more comparison with statistical analyzes.

2-Improve the size of Table 2 and 3.

3-Please reconfirm the description of Figure 2.

Thanks to the reviewer 3.

  1. Regarding the first comment, I increased the discussion section and the results section. Unfortunately, further comparisons by statistical method were of little value due to the sample size.
  2. I revised the tables by improving the display of the data.
  3. I revised the captions by correcting misprints

Round 2

Reviewer 2 Report

The authors have addressed all the questions. I recommend it for publication.